# Dynamic basis of lipopolysaccharide export by LptB$_2$FGC

**Marina Dajka[1], Tobias Rath[2], Nina Morgner[2], Benesh Joseph[1]***

[1]Department of Physics, Freie Universität Berlin, Berlin, Germany; [2]Institute of Physical and Theoretical Chemistry, Goethe Universität Frankfurt, Frankfurt, Germany

**Abstract** Lipopolysaccharides (LPS) confer resistance against harsh conditions, including antibiotics, in Gram-negative bacteria. The lipopolysaccharide transport (Lpt) complex, consisting of seven proteins (A-G), exports LPS across the cellular envelope. LptB$_2$FG forms an ATP-binding cassette transporter that transfers LPS to LptC. How LptB$_2$FG couples ATP binding and hydrolysis with LPS transport to LptC remains unclear. We observed the conformational heterogeneity of LptB$_2$FG and LptB$_2$FGC in micelles and/or proteoliposomes using pulsed dipolar electron spin resonance spectroscopy. Additionally, we monitored LPS binding and release using laser-induced liquid bead ion desorption mass spectrometry. The β-jellyroll domain of LptF stably interacts with the LptG and LptC β-jellyrolls in both the apo and vanadate-trapped states. ATP binding at the cytoplasmic side is allosterically coupled to the selective opening of the periplasmic LptF β-jellyroll domain. In LptB$_2$FG, ATP binding closes the nucleotide binding domains, causing a collapse of the first lateral gate as observed in structures. However, the second lateral gate, which forms the putative entry site for LPS, exhibits a heterogeneous conformation. LptC binding limits the flexibility of this gate to two conformations, likely representing the helix of LptC as either released from or inserted into the transmembrane domains. Our results reveal the regulation of the LPS entry gate through the dynamic behavior of the LptC transmembrane helix, while its β-jellyroll domain is anchored in the periplasm. This, combined with long-range ATP-dependent allosteric gating of the LptF β-jellyroll domain, may ensure efficient and unidirectional transport of LPS across the periplasm.

*For correspondence:
benesh.joseph@fu-berlin.de

Competing interest: The authors declare that no competing interests exist.

## eLife assessment

This study provides an **important** advance in the molecular understanding of the lipopolysaccharide export mechanism and machinery in bacteria. By using advanced spectroscopy approaches, the experiments provide **convincing** biophysical support for the dynamic behavior of the multisubunit Lpt transport system. This work has implications for understanding bacterial cell envelope biogenesis and developing drugs that target Gram negative pathogens.

## Introduction

Antibiotic resistance poses a critical threat to global health, necessitating a comprehensive understanding of bacterial defence mechanisms. The outer membrane (OM), a crucial component of Gram-negative bacteria, serves as an initial line of defence against antibiotics (***Nikaido, 2003***). Lipopolysaccharides (LPS) constitute a major component of the OM. LPS is a glycolipid consisting of lipid A, core oligosaccharides and an O-antigen (***Whitfield and Trent, 2014***). The LPS transport (Lpt) system plays a central role in the biogenesis of OM, contributing significantly to its impermeability and resilience against environmental stress and host defenses (***Wu et al., 2006***; ***Ruiz et al., 2008***; ***Chng et al., 2010***; ***Lundstedt et al., 2021***; ***Sperandeo et al., 2019***; ***Ho et al., 2018***). The pathways for LPS synthesis, transport and regulation offers potential targets for novel antibiotics (***Vetterli et al.,***

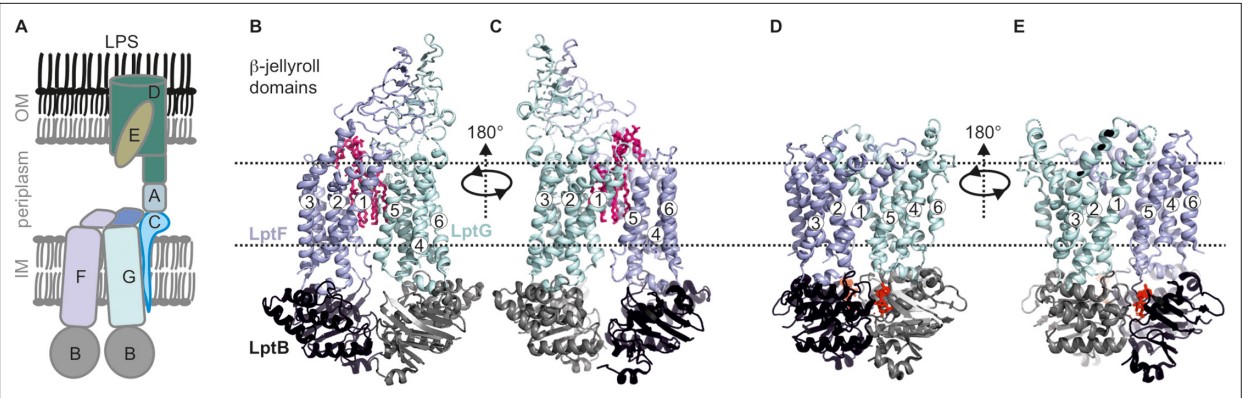

**Figure 1.** Lipopolysaccharide transport (Lpt) system. (**A**) the Lpt system consists of the LptB$_2$FG-C complex located in the inner membrane (IM). Periplasmic LptA connects this complex with LptDE, which is located in the outer membrane (OM). (**B, C**) Structures of LptB$_2$FG in the LPS-bound state (PDB ID: 6MHU) or (**D, E**) in the vanadate-trapped state (PDB ID: 6MHZ) are shown. In the vanadate-trapped structure, the cavity is collapsed with no space for LPS and the β-jellyroll domains are not resolved. The LPS molecule (magenta) and nucleotides (red) are highlighted.

The online version of this article includes the following figure supplement(s) for figure 1:

**Figure supplement 1.** LptB$_2$FG and LptB$_2$FGC structures in the apo and vanadate-trapped states.

2018; Zampaloni et al., 2024; Pahil et al., 2024; Mandler et al., 2018; Martin-Loeches et al., 2018). Comprising of seven essential components LptA-G (*Figure 1A*), the system orchestrates unidirectional translocation of LPS from the inner membrane (IM) to the OM (*Lundstedt et al., 2021*; *Törk et al., 2023*). The LptB$_2$FG complex forms an ATP-binding cassette (ABC) transporter in the IM (*Ruiz et al., 2008*; *Narita and Tokuda, 2009*; *Sperandeo et al., 2007*; *Thomas and Tampé, 2020*). The LptB subunits constitute the nucleotide-binding domains (NBD *Wang et al., 2014*) and interact in a head-to-tail manner to create two nucleotide binding sites (NBSs, for ATP or ADP). The F and G subunits together form the transmembrane domains (TMDs) to create the LPS binding pocket (*Dong et al., 2014*; *Tang et al., 2019*; *Dong et al., 2017*). The pseudo twofold symmetry of the TMDs creates two lateral gates on either side of the TMDs. LptC connects LptF-LptG (*Tang et al., 2019*; *Owens et al., 2019*; *Li et al., 2019*) with LptA (*Sherman et al., 2018*) to transfer LPS through the trans-periplasmic bridge towards LptDE (*Villa et al., 2013*; *Tran et al., 2010*; *Suits et al., 2008*; *Okuda et al., 2012*).

Previously, LptB$_2$FG apo structures revealed open NBDs with bound LPS inside the TMDs (*Figure 1B–C*; *Tang et al., 2019*; *Owens et al., 2019*; *Luo et al., 2017*). Upon vanadate trapping, the NBDs dimerize causing a collapse of the LPS binding pocket, which has been suggested as a post-translocation state (*Figure 1D–E*; *Tang et al., 2019*; *Li et al., 2019*). Presence of LptC enlarged the cavity leading to a weaker interaction with LPS (*Figure 1—figure supplement 1 A-B*; *Tang et al., 2019*; *Owens et al., 2019*; *Li et al., 2019*). In this conformation, the transmembrane (TM) helix of LptC weakly interacts with LptG, but forms extensive hydrophobic interactions with TM5 of LptF. The periplasmic β-jellyroll domains of LptF and LptG are very flexible leading to a reduced resolution. LptB$_2$FGC structures in the AMP-PNP or vanadate-trapped state did not reveal the transmembrane helix of LptC (TM-LptC), thereby revealing a nearly identical conformation of NBDs and TMDs as with the LptB$_2$FG structures (*Tang et al., 2019*; *Li et al., 2019*). Also, the β-jellyroll domains were barely resolved (*Figure 1—figure supplement 1 C-D*).

It was proposed that upon ATP binding, TM-LptC moves away from the LptG-TM1 – LptF-TM5 interface. This would lead to a collapse of the binding pocket and expulsion of LPS to the β-jellyroll domain of LptF. However, several details of this mechanism such as when and how TM-LptC dissociates and in what way that is transmitted to the interaction and dynamics of the β-jellyroll domains remain poorly understood. Likely, many intermediate states might exist between ATP binding and LPS translocation and the relevance of the available structures in a native-like lipid environment is unknown. Altogether, an integrated characterization of the intermediate structures and underlying dynamics is required to elucidate the functional mechanism. Here we characterized the conformational heterogeneity of LptB$_2$FG and LptB$_2$FGC using extensive pulsed dipolar electron spin resonance (ESR) spectroscopy experiments, which permits observations in micelles, lipid bilayers or even in the native environments for certain cases (*Galazzo and Bordignon, 2023*; *Tang et al., 2023*; *Yardeni*

*et al., 2019*; *Gopinath et al., 2024*; *Ketter et al., 2022*; *Gopinath and Joseph, 2022*; *Kugele et al., 2021*; *Galazzo et al., 2022*; *Bountra et al., 2017*; *Kapsalis et al., 2019*; *Ben-Ishay et al., 2024*; *Pierro et al., 2023*). These observations were further correlated with results obtained using laser-induced liquid bead ion desorption mass spectrometry (LILBID-MS; *Morgner and Robinson, 2012*; *Morgner et al., 2006*). Our results provide novel insights into the allosteric regulation of the structure and dynamics of the TMDs and the β-jellyroll domains during ATP binding and hydrolysis, which altogether drive LPS translocation.

## Results

### Conformational heterogeneity of the NBDs in LptB$_2$FG in micelles and liposomes

Initially, we characterized the dynamics of LptB$_2$FG in both DDM micelles and proteoliposomes (PLS) using pulsed electron-electron double resonance (PELDOR or DEER) spectroscopy (*Pannier et al., 2000*; *Schiemann et al., 2021*). This technique has been used for investigating several Type I, type II,

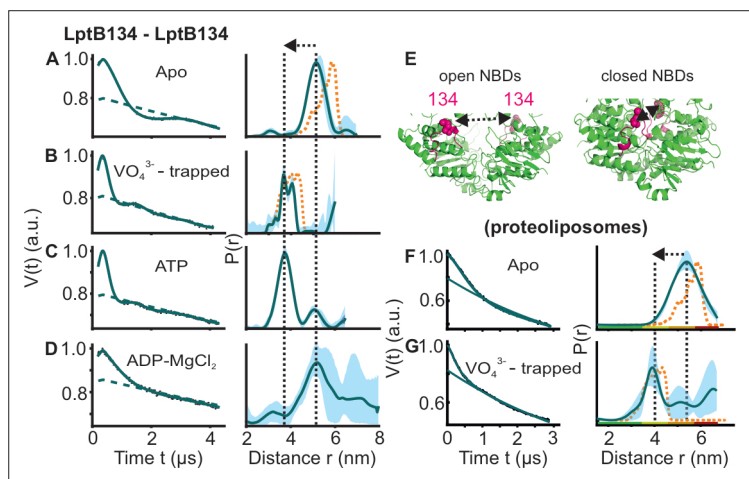

**Figure 2.** DEER/PELDOR data for the NBDs of LptB$_2$FG in micelles and PLS. (**A–D, F–G**) Primary data overlaid with the fits obtained using the DeerLab (*Fábregas Ibáñez et al., 2020*) program (A-D in micelles) or ComparitiveDeerAnalyzer (*Fábregas Ibáñez et al., 2020*; *Worswick et al., 2018* F-G in PLS) are shown in the left panels. The obtained distance distributions with a 95% confidence interval are shown on the right. Simulations for the open (PDB ID: 6MHU) or closed (PDB ID: 6MHZ) structures are overlaid with the apo or vanadate-trapped distances, respectively (in dotted orange line). (**E**) The spin labeled positions (as sphere) and the loops carrying them are highlighted (in red) on the open and closed structures. The PLS samples were analysed using ComparitiveDeerAnalyzer to account for the entire uncertainty including the partial dimensionality for spin distribution over the membrane surface. The color code for the distance distribution (**F–G**) relates the reliability for different features of the probability distribution with respect to the length of the observed dipolar evolution time. In the green zone, shape, width, and the mean distance are accurate. In the yellow zone, width and the mean, and in the orange zone, the mean distance are reliable. The arrow indicates the conformational change from the apo-state upon nucleotide binding.

The online version of this article includes the following source data and figure supplement(s) for figure 2:

**Source data 1.** Spin labeling efficiency for cysteine variants of LptB$_2$FG.

**Figure supplement 1.** Spin labeled positions in LptB$_2$FG.

**Figure supplement 2.** Size-exclusion chromatography (SEC) and SDS-PAGE for WT and spin labeled cysteine variants of LptB$_2$FG and LptB$_2$FGC.

**Figure supplement 2—source data 1.** Raw, unedited gel shown in *Figure 2—figure supplement 2*.

**Figure supplement 2—source data 2.** Uncropped, labelled gel shown in *Figure 2—figure supplement 2*.

**Figure supplement 3.** ATPase assay for spin labeled variants of LptB$_2$FG and LptB$_2$FGC.

**Figure supplement 4.** Room temperature continuous wave ESR spectroscopy of MTSL labeled variants in micelles.

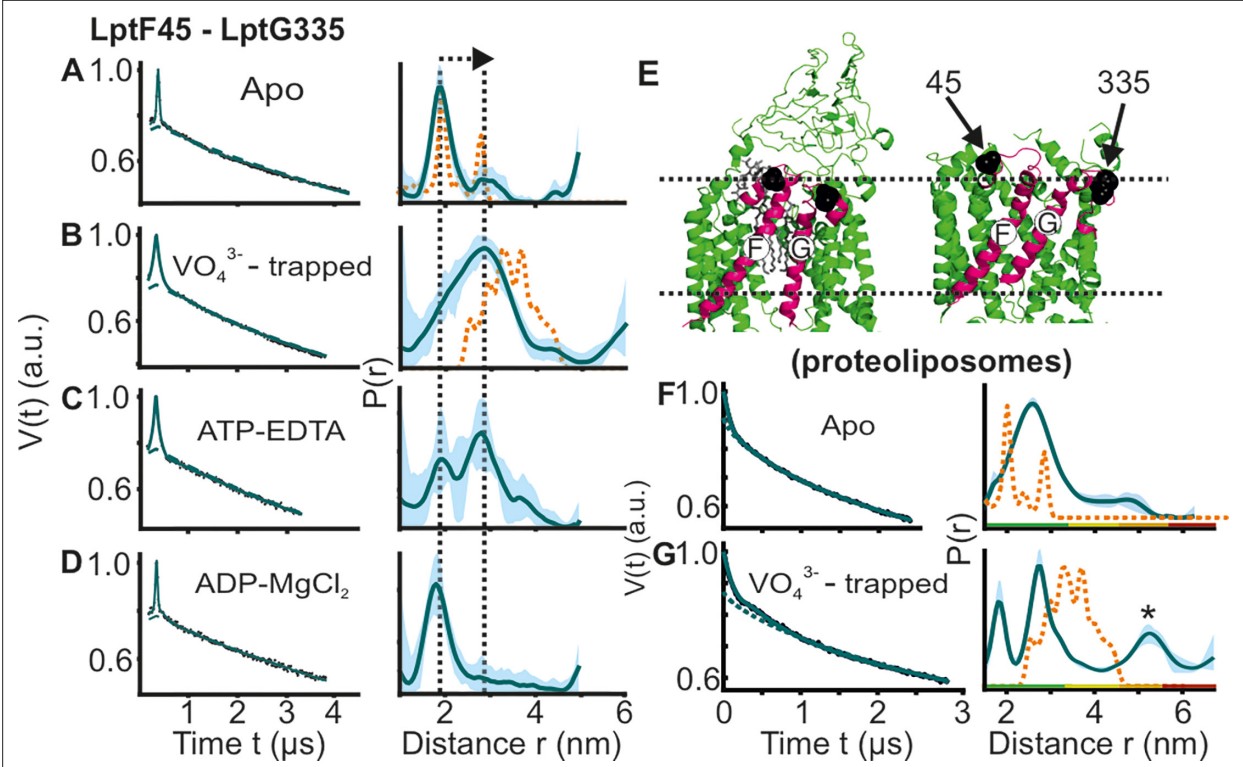

**Figure 3.** DEER/PELDOR data for the lateral gate-1 between LptF-TM1 – LptG-TM5 of LptB₂FG in micelles and PLS. Primary data overlaid with the fits obtained using the DeerLab program (A-D in micelles) or ComparitiveDeerAnalyzer (F-G in PLS) are shown in the left panels. The obtained distance distributions with a 95% confidence interval are shown on the right. (**E**) The spin labeled positions are highlighted on the open (PDB ID: 6MHU) and closed (PDB ID: 6MHZ) structures and the corresponding simulations are overlaid with the apo or vanadate-trapped distances, respectively (in dotted orange line). The distance peak indicated with an asterisk in G is not interpreted for the reason of a larger uncertainty. The color code for the distance distribution (**F–G**) corresponds to the description as given in **Figure 2**. The arrow indicates the conformational change from the apo-state upon nucleotide binding.

type III, and type IV ABC transporters (**Tang et al., 2023**; **Bountra et al., 2017**; **Grote et al., 2009**; **Joseph et al., 2011**; **Barth et al., 2018**; **Verhalen et al., 2017**; **Borbat et al., 2007**; **Hutter et al., 2019**; **Majsnerowska et al., 2013**). LptB₂FG belongs to the type VI class of ABC transporter. For this purpose, we engineered spin pairs along the NBDs, TMDs and the β-jellyroll domains (**Figures 2–6**, **Figure 2—figure supplement 1**). Following reaction with a cysteine, the (1-Oxyl-2,2,5,5-tetramethyl-3-pyrroline-3-methyl methanethiosulfonate, MTSL) label creates the side chain denoted as R1 (**Hubbell et al., 1998**). The functionality of the spin labeled variants after size-exclusion chromatography was confirmed using ATP-ase activity assay (**Figure 2—figure supplement 2**, **Figure 2—figure supplement 3**). All the cysteine variants could be labeled with high efficiency (**Figure 2—figure supplement 4** and **Figure 2—source data 1**). We investigated LptB₂FG under several conditions including the apo-state or after incubating with ATP-EDTA (EDTA to chelate divalent metal ions and to inhibit any residual ATPase activity, denoted as ATP), ATP-Mg²⁺-VO³⁻4 (vanadate-trapped transition state) or ADP-Mg²⁺ (post-hydrolysis state).

In agreement with the apo structure, the NBDs displayed a mean interspin distance centred around 5 nm in micelles when observed at position B_M134R1 (**Figure 2A–D**). This distance decreased below 4 nm in the vanadate-trapped state. With ATP a mixed population was observed with the major peak corresponding to the closed conformation. Thus, Mg²⁺ ions are not necessary for the closure of the NBDs. With ADP-Mg²⁺ the NBDs do not close, though the overall flexibility is somewhat increased (as gauged from the width of the distribution). We performed additional experiments after reconstitution into PLS (**Figure 2F–G**). Overall, the data in the apo-state revealed a similar mean distance, yet with a larger width suggesting an increased dynamics. As observed in micelles, vanadate-trapping closed the NBDs in PLS as well. Overall, the nucleotide-induced closure as observed in micelles (and the structures) is maintained in the native-like lipid bilayers for the NBDs.

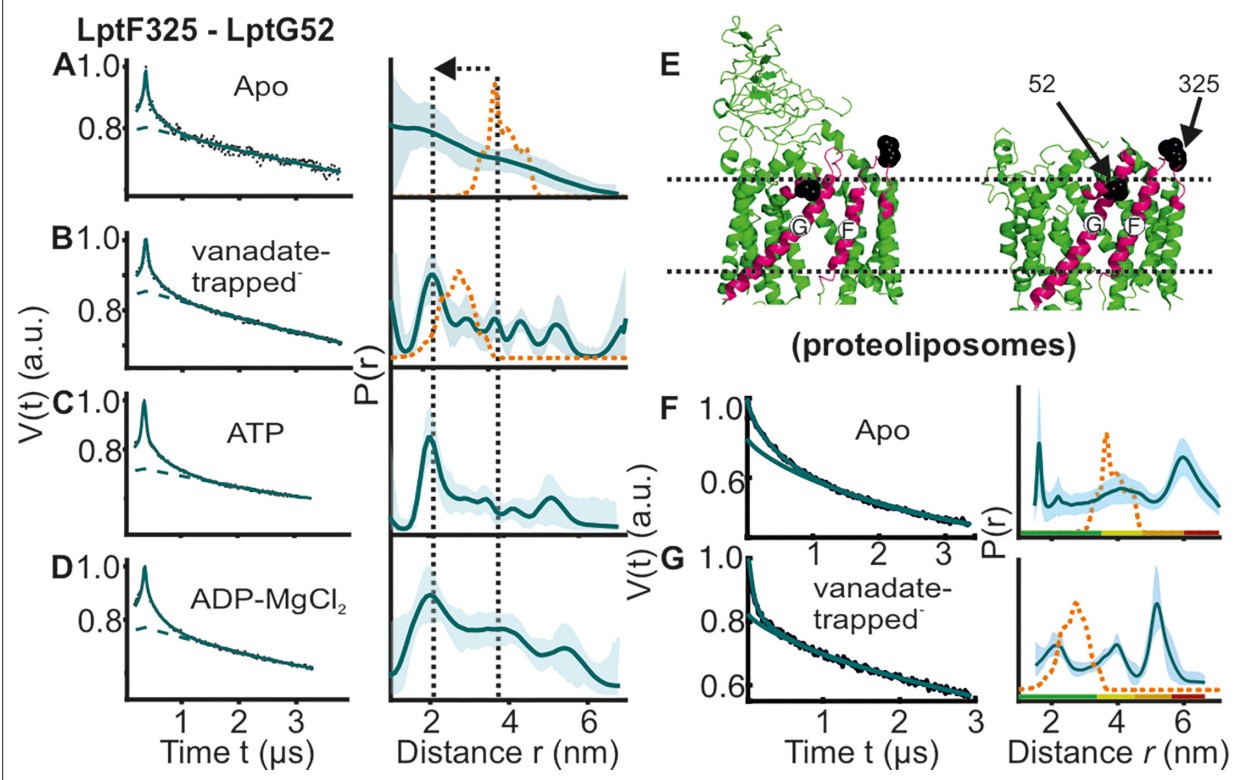

**Figure 4.** DEER/PELDOR data for the lateral gate-2 between LptF-TM5 – LptG-TM1 of LptB$_2$FG in micelles and PLS. Primary data overlaid with the fits obtained using the DeerLab program (A-D in micelles) and DeerNet (**F**) or ComparitiveDeerAnalyzer (**G**) in PLS are shown in the left panels. The obtained distance distributions with a 95% confidence interval are shown on the right. Due to a larger uncertainty arising from the broad distribution and the limited time-window, output from DeerNet is shown in F. (**E**) The spin labeled positions are highlighted on the open (PDB ID: 6MHU) and closed (PDB ID: 6MHZ) structures and the corresponding simulations are overlaid with the apo or vanadate-trapped distances, respectively (in dotted orange line). The color code for the distance distribution (**F–G**) corresponds to the description as given in *Figure 2*. The arrow indicates the conformational change from the apo-state upon nucleotide binding.

The online version of this article includes the following figure supplement(s) for figure 4:

**Figure supplement 1.** Conformation of the loop carrying the spin labeled position 325 in LptF at the second lateral gate.

**Figure supplement 2.** Comparison of room temperature continuous wave ESR spectroscopy of MTSL labeled variants between DDM micelles and proteoliposomes (PLS).

## Conformational heterogeneity of the lateral gate-1 of LptB$_2$FG in micelles and PLS

We introduced two spin pairs to independently monitor the lateral gates of the TMDs. The lateral gate formed by LptF-TM1 – LptG-TM5 and LptG-TM1 – LptF-TM5 were monitored using F_A45R1 – G-I325R1 and F_L325R1 – G-A52R1 pairs, respectively. For the apo-state in micelles, the distances for the lateral gate-1 are similar to the corresponding simulation on the LPS-bound structure (*Figure 3A*). Vanadate-trapping resulted in an increase of the distances in agreement with the simulation. Though the central cavity collapses, this increased distance results from a coordinated movement of LptF-TM1 towards the cavity accompanied with an outward shift of LptG-TM5 (*Figure 3E*, *Figure 1—figure supplement 1* and *Figure 2—figure supplement 1*). Addition of ATP resulted in an equilibrium between the two conformations. Altogether, the PELDOR data validate a LPS binding competent conformation in the apo- and ADP-Mg$^{2+}$ states accompanied with a collapse of the gate in the ATP and vanadate-trapped post-hydrolytic state. Interestingly, for the apo state in PLS, this gate exhibited a broader distribution spanning the range corresponding to both structures (*Figure 3F–G*). In the vanadate-trapped state the overall distribution got narrowed (the minor peak at a longer distance is not interpreted due to a larger uncertainty). However, the overall spread of the distribution remained

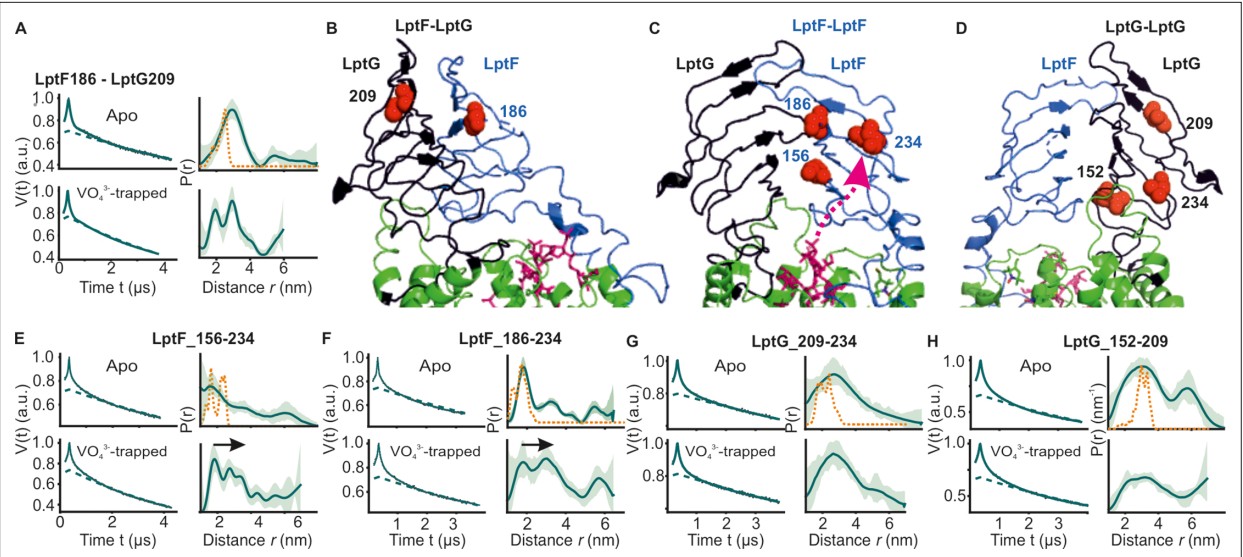

**Figure 5.** DEER/PELDOR data for β-jellyroll domains of LptB₂FG. (**A**) Primary data for LptF-LptG β-jellyrolls overlaid with the fits obtained using the DeerLab (*Fábregas Ibáñez et al., 2020*) program are shown in the left panels. The obtained distance distributions with a 95% confidence interval are shown on the right. (**B–D**) The spin labeled positions are highlighted on the LptB₂FG apo structure and the bound LPS is shown in stick representation (PDB ID: 6MHU). (**E, F**) Primary data overlaid with the fits and the corresponding distance distributions (right) from PELDOR experiments within the β-jellyroll domain of LptF or LptG (**G, H**). Corresponding simulations (where structures are available having the positions resolved) are overlaid (in orange).

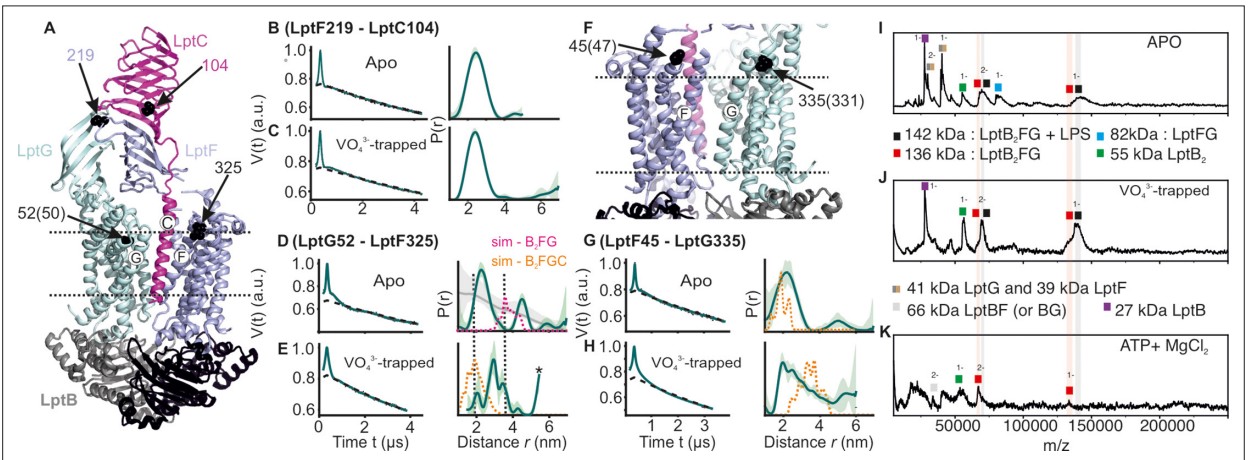

**Figure 6.** DEER/PELDOR data for the lateral gates of LptB₂FGC and LILBID-MS. (**A**) The LptB₂FGC structure (PDB ID: 6MJP) from *Vibrio cholerae*. The helices from LptF and LptG forming the lateral gate-2 is shown and the positions analogous to those we investigated in the *E. coli* structure are highlighted (in sphere representation and indicated with an arrow, when the amino acid number is different, it is indicated inside the bracket). Similarly, the lateral gate-1 is highlighted in panel F. (**B–E, G–H**) Primary PELDOR data overlaid with the fits obtained using the DeerLab (*Fábregas Ibáñez et al., 2020*) are shown in the left panels. The obtained distance distributions with a 95% confidence interval are shown on the right. The corresponding distances for LptB₂G is overlaid for a comparison (in grey for the apo state) in panel D and asterisk in panel E indicate unreliable distances. Simulation on the LptB₂FGC apo structure (in orange) is overlaid. In panel D, corresponding distances for LptB₂FG is overlaid (in grey) and the simulation for LptB₂FG structure is shown (in magenta) as position L325 is not resolved in the *E. coli* LptB₂FGC structure. (**I–K**) LILBID-MS data for detergent solubilized LptB₂FG in different states as indicated.

similar, suggesting a more flexible conformation of this gate in PLS, which is minimally affected by ATP binding.

## The putative lateral gate-2 for LPS entry exhibits a large conformational heterogeneity

Interestingly, the lateral gate-2 between LptG-TM1 – LptF-TM5 revealed a more dynamic behavior in micelles (*Figure 4A–D*). In the apo state, it has a broad conformational distribution spanning the LPS bound (open) and vanadate-trapped (closed) structures (corresponding simulations are shown in orange lines). In the vanadate-trapped and ATP samples, the major population is centred at 2 nm (which is closer to the simulation on the vanadate-trapped structure). The decreased interspin distances arise from a collapse of the cavity upon vanadate-trapping (*Figure 4E*, *Figure 1—figure supplement 1* and *Figure 2—figure supplement 1*). ADP-Mg$^{2+}$ also gave a broad distribution comparable to the apo-state. Thus, in the apo-state this gate appears to exist in an equilibrium between the two conformations as observed from the corresponding structures. ATP binding or vanadate-trapping shifts the equilibrium towards the collapsed conformation. In PLS as well, the apo state revealed a broad distribution validating the heterogeneity in the membrane environment (*Figure 4F–G*). Vanadate-trapping somewhat narrowed the distribution with (only) a fraction of the distances overlaying with the simulation for the corresponding structure (and with a more pronounced heterogeneity as compared with the micellar sample, *Figure 4B*). Position A52 on LptG is located at the beginning of the neighboring TM2. Position L325 is located on the short loop between TM5 and TM6 in LptF. This loop is resolved with clear density and has a similar orientation in the apo and vanadate-trapped structures with minimum deviation (*Figure 4E*, *Figure 4—figure supplement 1*). Further, the observed heterogeneity is distinctly modulated upon LptC binding (*Figure 6D–E*), suggesting that the internal flexibility around the labeled sites might have the least contribution to the broad distribution we experimentally observed. Confirming these observations, the room temperature continuous wave ESR spectra revealed the least flexibility for this spin pair (*Figure 2—figure supplement 4* and *Figure 4—figure supplement 2*). Comparing this heterogeneity from the PELDOR data (*Figure 4*) with the corresponding simulations (in orange, *Figure 4A–B*), it appears that the structures captured two of the states from the broad conformational space. This gate is suggested to be the entry point for LPS (*Owens et al., 2019*) and the enhanced dynamics we observed might be required for efficient interaction with LPS and LptC (please see later sections). Supporting these observations, LptG-TM1 and LptF-TM5 helices interact rather weak in the structures, in particular in the LPS-bound state (*Figure 1B–C*, *Figure 1—figure supplement 1* and *Figure 2—figure supplement 1*).

## Conformational dynamics of the β-jellyroll domains

We further investigated the interaction between the β-jellyroll domains in LptB$_2$FG. Distances between positions F_S186R1 and G_V209R1 located on these domains revealed a major peak in agreement with simulation on the LPS-bound structure (*Figure 5A–B*) and a previous study (*Cina and Klug, 2024*). The overall spread of the distribution remained rather similar in the vanadate-trapped state revealing a stable interaction. We also probed the internal flexibility of LptF β-jellyroll using F_S156R1– I234R1 and F_S186R1 – I234R1 pairs, which are located on the putative entry gate loops for LPS (*Owens et al., 2019*; *Figure 5C and E–F*). The LptF β-jellyroll in the apo-state showed a broad distribution, but with a major population staying in the closed conformation as observed in the structure (*Figure 5E and F*). Interestingly, in the vanadate-trapped state, the overall distribution shifted towards longer distances revealing an opening of the LPS entry gate. Altogether, these observations reveal a long-range allosteric coupling between nucleotide binding at the NBDs in the cytoplasm and opening of the LPS entry gate of LptF β-jellyroll in the periplasm. Such a tight coupling might prime this domain to receive LPS once it is released from the cavity inside the TMDs. A strikingly different response was observed for the β-jellyroll domain of LptG. The equivalent spin pairs G_V209R1 – G_L234R1 and G_L152R1 – V209R1 revealed a considerably enhanced flexibility independent of ATP (*Figure 5D and G–H*) and revealing no such allosteric coupling with the NBDs.

## Conformational heterogeneity of the the β-jellyroll domains and lateral gates in LptB₂FGC

Next, we investigated how LptC modulates dynamics of the lateral gates and the β-jellyroll domains in LptB₂FG. Measurements between the β-jellyroll domains of LptF and LptC (F_L219R1 – C_T104R1) revealed a narrow distance distribution and hence a stable interaction between them in both apo and vanadate-trapped states (*Figure 6A–C*). At the lateral gate-1 (F_A45R1 – G_I335R1, *Figure 6F and G–H*), the distance in the apo state is similar to the simulation on the structure. In the vanadate trapped state, the spread of the distance distribution is increased, spanning the range of the simulations corresponding to both apo and vanadate-trapped structures (in the latter structure, TM-LptC is absent at the gate leading to a collapse of the cavity and an identical conformation with LptB₂FG, though the distances increase due to an outward movement of LptG-TM5, see *Figure 3B and G*, *Figure 1—figure supplement 1* and *Figure 2—figure supplement 1*). Distances corresponding to the apo state are present possibly due to an incomplete vanadate-trapping (or an equilibrium between the two conformations, which is rather unlikely) for this sample. Overall, the results support TM-LptC dissociation in the vanadate-trapped state as observed in the structure.

A distinct response is observed for the lateral gate-2 (F_L325R1 – G_A52R1, *Figure 6A and D–E*). LptC binding reduced the overall flexibility of this gate into two defined distance peaks as compared to LptB₂FG alone (*Figure 4A* vs. *Figure 6D*). LptC binding is shown to increase the separation between these helices (*Figure 1—figure supplement 1A*). The major distance peak (centred around 2 nm) is shorter than the simulation on the LptB₂FG apo structure (in magenta, *Figure 6D*, top panel), which therefore likely represents a conformation in which TM-LptC is released. The second peak corresponds to a larger separation between these helices, possibly representing the structure having the TM-LptC inserted at the gate (as position L325 is not resolved in the *E. coli* structure, corresponding simulation is not presented). The first conformation appears to be more favored under the experimental conditions. In the vanadate-trapped state, the second peak disappeared and the overall distribution centred towards the TM-LptC released (LptB₂FG) conformation (*Figure 6E*). This is in line with the lack of density for TM-LptC in the vanadate-trapped LptB₂FGC structure. The experimental distribution is longer than the corresponding simulation (orange line in *Figure 6E*), suggesting a somewhat farther separation between these helices.

We further probed LPS release in LptB₂FG using LILBID-MS (*Figure 6I–K*). In the apo and vanadate-trapped states, the majority of the observed complex peaks represent LPS bound to LptB₂FG, while some LPS-free LptB₂FG can as well be seen (indicated in grey and red shades, respectively for both 1- and 2- charge states). Thus, LPS is co-purified with LptB₂FG in micelles. Presence of LPS-bound LptB₂FG in the vanadate-trapped state is not unexpected as the PELDOR data revealed a heterogenous conformation for the LPS entry gate other than the collapsed conformation (*Figure 4B and G*). However, under the hydrolysing conditions, LPS is completely released from LptB₂FG.

## Discussion

LPS transport by LptB₂FG has remained elusive for many reasons. LptB₂FG and LptB₂FGC structures revealed a similar structure in the vanadate-trapped state leaving no trace for TM-LptC and the β-jellyroll domains were not resolved (*Figure 1—figure supplement 1D*). Available structures possess either AMP-PNP or ADP-VO₄³⁻ as nucleotides (both in a closed conformation of the TMDs and NBDs), thereby limiting details on how ATP binding and hydrolysis are coupled to the conformational changes and LPS transport.

For the NBDs and the two lateral gates of LptB₂FG, PELDOR data validated the conformational changes in micelles as observed from the structures. Upon ATP binding and/or vanadate-trapping, the NBDs and the two lateral gates moved in a coordinated manner to collapse the LPS binding pocket (*Figures 2–4*). However, the lateral gate-2 forming the LPS entry site (TM1G -TM5F monitored by F_L325 – G_A52, *Figure 4*, *Figure 1—figure supplement 1* and *Figure 2—figure supplement 1*) exhibits a flexible conformation, which might facilitate interaction with LPS and or LptC. In agreement, LptC binding restricted the overall flexibility of this gate into two distinct conformations corresponding to a collapsed and an open conformation in which TM-LptC is likely inserted into the TMDs (*Figure 6D*). The minor peak at longer distances might corresponds to a conformation in which TM-LptC and LPS can enter the TMDs while the β-jellyroll domains of LptC and LptF interact. The minimal interaction

between the corresponding helices (*Figure 1—figure supplement 1* and *Figure 2—figure supplement 1*) in the LptB$_2$FG structures suggest that the flexibility is inbuilt and might have an important role for the function. The PLS environment modulates the observed conformation in LptB$_2$FG. The lateral gate-1 has a broader distribution in PLS, which is minimally affected by vanadate-trapping (*Figure 3F–G*). Also, the heterogeneity for the lateral gate-2 is more pronounced in PLS. The liposomes are made from *E. coli* polar lipid extract. In the polar lipid extract, phosphatidylethanolamine is the predominant lipid component with minor amounts of phosphatidylglycerol and cardiolipin. Thus, the differences in the heterogeneity we observed in proteoliposomes might not be due to the presence of LPS. Overall, the observations in PLS are qualitatively similar to the micellar sample (also see *Figure 4—figure supplement 2* for the continuous wave (cw) ESR spectra) and further experiments are required to clarify how the membrane would influence LptB$_2$FGC conformation.

In LptB$_2$FG, the two β-jellyroll domains stably interact in the apo and vanadate-trapped conditions (*Figure 5*). Interestingly, binding of nucleotides is allosterically coupled to a selective opening of LptF β-jellyroll with little effect on the LptG β-jellyroll (*Figure 5E–H*). This opening was shown to be essential for cell growth and it was suggested that movement of LPS, but not ATP binding or hydrolysis pushes the gate open (*Owens et al., 2019*). However, our results confirm that the gate opening is allosterically regulated through ATP binding and/or hydrolysis. This β-jellyroll start from TM3 and ends on TM4. TM3 is connected to the coupling helix and TM4 is connected to the cytoplasmic loop-2, which harbours the conserved R292 directly interacting with the nucleotide (*Tang et al., 2019*). It is likely that one or both of these motifs are involved in the allosteric communication. Notably, the LptG β-jellyroll domain exhibits significant internal flexibility (*Figure 5G-H*). Biochemical studies showed that conserved residues in this domain are essential for cell growth (*Tang et al., 2019*). Although there is little evidence for a direct role of this domain in LPS binding, the observed dynamics might be important for efficient transport. Similarly, in LptB$_2$FG-C, the β-jellyrolls of LptC and LptF interact both in both apo and vanadate-trapped states (*Figure 6A–C*), which altogether might form a continuous pathway for LPS transfer towards LptA. The β-jellyrolls were not fully resolved in the vanadate-trapped structure of LptB$_2$FG (*Figure 1*) and LptB$_2$FGC (*Li et al., 2019*), revealing a considerable flexibility. The exact role for this enhanced dynamics is unclear and it may somehow facilitate LPS release. The narrow distance distributions we observed reveal that these domains might move as rigid bodies while sampling a broad conformational space.

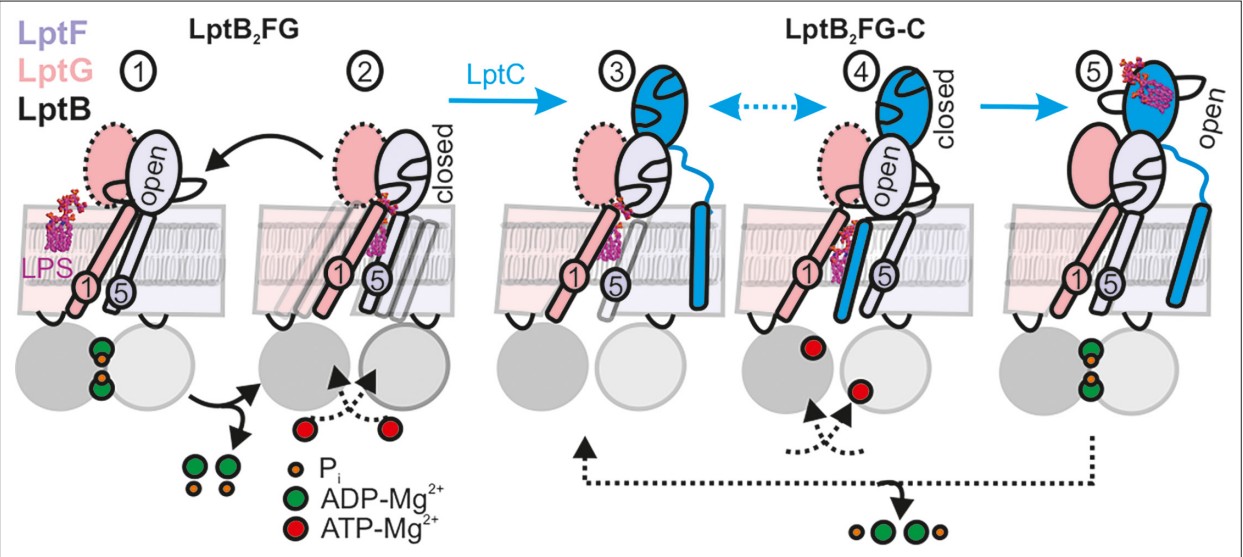

**Figure 7.** LPS translocation mechanism. (1-2) The lateral gate-2 formed by LptG-TM1 and LptF-TM5 in LptB$_2$FG, which is dynamic in the apo state (2) is shown (see *Figure 4*). ATP binding, which leads to hydrolysis (1) collapses the cavity and releases LPS. (3,4) Interaction with LptC limits heterogeneity, creating an equilibrium between TM-LptC inserted and released conformations (*Figure 6D*). ATP binding opens the periplasmic gate in LptF β-jellyroll (4) and subsequent closure of the NBDs leads to the collapse of the cavity and LPS transfer. As LPS moves to LptC, LptF β-jellyroll might close accompanied with an opening of LptC β-jellyroll (*Tang et al., 2019*) (5). Dissociation of ADP-Mg$^{2+}$ will initiate the next cycle according to the Pez mechanism. Dotted line for LptG β-jellyroll indicates enhanced inter-domain dynamics (see *Figure 5G–H*).

The LILBID-MS data show that LPS is released upon ATP hydrolysis by LptB$_2$FG (*Figure 6I–K*, *Figure 7* steps 1–2). Thus, LptC is required for a productive transport cycle in the wild-type protein (*Sherman et al., 2018*; *Falchi et al., 2023*; *Wilson and Ruiz, 2022*). Our observations suggest that the TM-LptC dynamically associates and dissociates at the lateral gate-2, even in the apo-state while its β-jellyroll domain is firmly bound to LptF β-jellyroll (*Figure 7*, step 3). Such a dynamic behavior of TM-LptC has previously been suggested based on the *Klebsiella pneumoniae* LptB$_2$FGC structure, which showed density for the short N-terminus segment of TM-LptC (*Luo et al., 2021*). Vanadate trapping induces the TM-LptC released conformation of the lateral gates (*Figure 6D–E and G–H*), in agreement with the structures, which also leads to LPS release (*Tang et al., 2019*; *Owens et al., 2019*). Thus, ATP binding/hydrolysis might be coupled to the collapse of the TMDs and expulsion of LPS. Inferring from our observations on LptB$_2$FG (*Figure 5*), ATP binding might allosterically open the LptF β-jellyroll (*Figure 7*, step 4), which might subsequently close after passing LPS to the LptC β-jellyroll to prevent any backflow (*Tang et al., 2019*; *Figure 7*, step 5). Through repeating this cycle, LptB$_2$FG might push LPS along the trans-periplasmic bridge in a unidirectional manner according to the Pez candy dispenser model (*Okuda et al., 2016*). The possibility to observe LptB$_2$FG and LptB$_2$FGC offers further opportunity to characterize the dynamics of the β-jellyroll domains, LptB$_2$FG – TM-LptC interaction as well as other intermediate states of the transport cycle even in the native-like lipid bilayers. These experiments, which are beyond the current scope of the study are in progress in our laboratory. As the persistently interacting, yet dynamic LptB$_2$FGC complex forms the functional LPS transport unit in the inner membrane, a detailed understanding of its dynamics would facilitate development of novel drugs against Gram-negative bacterial pathogens.

# Materials and methods

## Expression, purification and spin labeling of LptB$_2$FG and LptB$_2$FGC

The plasmid pETDuet-*lptB-lptFG*, including a C-terminal His-tag on LptB, was used to transform in *E. coli* BL21(DE3) C43 for LptB$_2$FG expression. For LptB$_2$FGC a co-expression with the pCDFDuet-1-*lptC* together with pETDuet-*lptB-lptFG* was performed. For bacterial cell growth Luria broth (LB) medium with 100 µg/mL ampicillin (for LptB$_2$FG, and additionally 50 µg/mL streptomycin for LptB$_2$FGC) was incubated at 37 °C until an optical density of 1.0 at 600 nm. The protein expression was induced with 0.2 mM or 0.4 mM isopropyl β-D-1-thiogalactopyranoside (IPTG) for LptB$_2$FG and LptB$_2$FGC, respectively. After overnight expression at 18 °C the cells were collected by centrifugation. Cell pellets were resuspended in lysis buffer (20 mM HEPES, pH 7.5, 300 mM NaCl) with a homogenizer and lysed by French press with an additional spatula tip of DNAse I, lysozyme and 100 µg/ml PMSF. Cell debris was removed by centrifugation at 12,000x *g* for 20 min at 4 °C. The membranes were pelleted by ultra-centrifugation at 200,000 x *g* for 2 hr at 4 °C. Membranes were flash-frozen in liquid nitrogen and stored at –80 °C. For solubilization lysis buffer with 1% (w/v), DDM was used and after homogenization membranes were incubated for 30 min at 4 °C. The protein solution was ultra-centrifuged at 100,000 x *g* for 35 min at 4 °C. The supernatant was incubated with washed Ni Sepharose (High Performance, GE Healthcare) for 45 min at 4 °C and loaded on an empty PD-10 column (GE Healthcare). The column was washed with 15 column volumes of wash buffer containing 20 mM HEPES, pH 7.5, 300 mM NaCl, 0.05% (w/v) DDM, 5% (v/v) glycerol and 40 mM imidazol. The protein bound to the Ni Sepharose was labeled with 2 column volumes of 20 mM HEPES, pH 7.5, 300 mM NaCl, 0.05% (w/v) DDM, 5% (v/v) glycerol and 300 µM MTSL (1-Oxyl-2,2,5,5-tetramethyl-3-pyrroline-3-methyl)methanethiosulfonate for 1.5 h at 4 °C. The column was washed with 40 column volumes of the same buffer (without MTSL) and further eluted with 300 mM imidazole and immediately deslated using a PD-10 desalting column (GE Healthcare) into the same buffer. The protein was then concentrated to about 100 µM (Amicon Ultra-15, PLQK Ultracel-PL Membrane, 50 kDa, Merck, Millipore). After size-exclusion chromatography with a Superdex 200 Increase10/300 GL column (GE Healthcare), the protein fractions were collected and concentrated to ~40 µM protein. For PELDOR spectroscopy different samples were prepared. For the ATP-EDTA sample, 5 mM ATP and 0.5 mM EDTA were added (5 min, 37 °C). The ADP-MgCl$_2$ sample contained 5 mM ADP and 5 mM MgCl$_2$ (1 min, RT). The vanadate-trapped sample was prepared with 5 mM ATP, 5 mM MgCl$_2$ and 5 mM *ortho*-vanadate and the sample was incubated for 5–10 minutes at 37 °C. All the samples were adjusted to the same final protein and DDM concentration (0.05%).

## Proteoliposome reconstitution and PELDOR sample preparation

Liposomes (20 mg/mL, 20 mM HEPES, 300 mM NaCl, 10% (v/v) glycerol, pH 7.5) made from *E. coli* polar lipid extract (Avanti) were mixed with 0.14% Triton X-100 and incubated for 30 min at room temperature. Liposomes and protein (in the same buffer additionally containing 5% glycerol and 0,05% (w/v) DDM) were mixed at 1:10 (w/w) ratio to a final lipid concentration of 4 mg/mL in 10 mL. The protein-lipid suspension was gently incubated for 1 hr at room temperature. Afterwards Bio-Beads SM-2 (Bio-Rad) was added and was gently mixed for 30 min at room temperature and moved to 4 °C. The Bio-Beads were added three more times for the following incubation periods at 4 °C: 1 hr, overnight and 2 hr. Proteoliposomes were diluted to 2% (v/v) glycerol in 20 mM HEPES, 300 mM NaCl, pH 7.5 and were pelleted at 200,000x *g* (4 °C) and resuspended into the same buffer. PELDOR samples were prepared as described above for DDM micelles with an additional freeze-thaw (5 x) cycle. Samples were frozen in liquid nitrogen and stored at –80 °C.

## ATP assay

The ATP assay was performed according to a protocol modified from *Morbach et al., 1993*. Five µg of purified LptB$_2$FG or LptB$_2$FGC were incubated at 37 °C for 8 min in 25 µL reaction volume on a 96-well plate with different ATP concentrations (0, 0.125, 0.25, 0.5, 1, 2, 3, and 5 mM) in 20 mM HEPES pH 7.5, 150 mM NaCl, 0.05% (w/v) DDM and 5 mM MgCl$_2$. The reaction was stopped with addition of 150 µL of 20 mM H$_2$SO$_4$. A 50 µL working solution [50 ml of ddH$_2$O, 10 ml of ≥95% H$_2$SO$_4$ and 73.4 mg of malachite green chloride, 40 µl of 11% Tween 20, and 500 of µl 7.5% (w/v) (NH$_4$)$_6$Mo$_7$O$_{24}$.4H$_2$O solution] was added and incubated for 8 min at RT and the absorbance was measured at 620 nm in a microplate reader. The data was analyzed using Origin 2018.

## Continuous wave (cw) ESR spectroscopy

Continuous wave ESR spectroscopy measurements were performed on a X-band Bruker EMXnano benchtop spectrometer. A 16 µL of the protein sample was measured in a 0.64 mm diameter micropipette (BRAND, Germany) with 100 kHz modulation frequency, 0.6–2 mW microwave power, 0.15 mT modulation amplitude and 18 mT sweep width.

## Pulsed electron-electron double resonance spectroscopy

DEER/PELDOR experiments were conducted on a Bruker Elexsys E580 Q-Band (34 GHz) pulsed ESR spectrometer equipped with an arbitrary waveform generator (SpinJet AWG, Bruker), a 50 W solid-state amplifier, a continuous-flow helium cryostat, and a temperature control system (Oxford Instruments). Measurements were carried out at 50 K using a 10–20 µL frozen sample containing 15–20% glycerol-d$_8$ in a 1.6 mm quartz ESR tube (Suprasil, Wilmad LabGlass) with a Bruker EN5107D2 dielectric resonator. The phase memory time ($T_M$) measurements were performed with a 48 ns π/2–τ–π Gaussian pulse sequence with a two-step phase cycling after incrementing $\tau$ in 4 ns steps. A dead-time free four-pulse sequence with a 16-step phase cycling (x[x][x$_p$]x) was used for DEER measurements (*Tait and Stoll, 2016*). A 38 ns Gaussian pump pulse (with a full width at half maximum (FWHM) of 16.1 ns) was employed, along with a 48 ns observer pulse (FWHM of 20.4 ns *Teucher and Bordignon, 2018*). The pump pulse was placed at the maximum of the echo-detected field swept spectrum, and the observer pulses were set 80 MHz lower. Deuterium modulations were averaged by progressively increasing the first interpulse delay by 16 ns over 8 steps. The data analysis was performed with the ComparativeDeerAnalyzer 2.0 (CDA *Fábregas Ibáñez et al., 2020*; *Worswick et al., 2018*) or the DeerLab program. The distance distributions were simulated on the structures (PDB 6MHU, 6MHX, 6MI7, and 6MI8) using a rotamer library approach as implemented in the MATLAB-based MMM2022.2 software package (*Jeschke, 2021*).

## Liquid bead ion desorption mass spectrometry (LILBID-MS)

For analysis by LILBID-MS, samples were buffer exchanged to 20 mM Tris(HCl), 0.05% DDM, pH 7.5. For ATP-Mg$^{2+}$ sample, 2 mM ATP and 0.5 mM MgCl$_2$ were added prior to the buffer exchange and the sample was incubated for 1 min at room temperature. After dilution to 10 µM protein, 4 µL of the sample were loaded directly into a droplet generator (MD-K-130, Microdrop Technologies GmbH, Germany). This piezo driven device generates droplets of a diameter of around 50 µm at a frequency of 10 Hz. The droplets are then transferred into vacuum, and irradiated by an IR laser working at a

wavelength of 2.8 μm having a maximum energy output of 23 mJ per pulse (6 ns). The free ions were accelerated into a homebuilt time of flight analyser by applying an acceleration voltage in the Wiley McLaren type ion optics. The voltage between the first and the second lens was set to –4.0 kV. At 5–25 μs after irradiation, the first lens was pulsed to –6.6 kV for 370 μs, while the reflectron worked at –7.2 kV. Detection was conducted by a homebuilt Daly type detector, optimized for high m/z.

## Acknowledgements

This work was financially supported from the Deutsche Forschungsgemeinschaft via the Emmy Noether program (JO 1428/1–1), SFB 1507 – 'Membrane-associated Protein Assemblies, Machineries, and Supercomplexes', and a large equipment fund (438280639) to BJ. NM acknowledges funding by the Deutsche Forschungsgemeinschaft (DFG, German Research Foundation) —Project-ID 426191805. BJ and MD thanks Jingyi Liu for establishing LptB$_2$FGC purification and PELDOR experiments.

## Additional information

### Funding

| Funder | Grant reference number | Author |
|---|---|---|
| Deutsche Forschungsgemeinschaft | JO 1428/1–1 | Benesh Joseph |
| Deutsche Forschungsgemeinschaft | 438280639 | Benesh Joseph |
| Deutsche Forschungsgemeinschaft | SFB 1507 | Benesh Joseph |
| Deutsche Forschungsgemeinschaft | 426191805 | Nina Morgner |

The funders had no role in study design, data collection and interpretation, or the decision to submit the work for publication.

### Author contributions

Marina Dajka, Conceptualization, Data curation, Formal analysis, Validation, Investigation, Visualization, Methodology, Writing - review and editing; Tobias Rath, Nina Morgner, Data curation, Formal analysis, Validation, Visualization, Methodology, Writing - review and editing; Benesh Joseph, Conceptualization, Resources, Formal analysis, Supervision, Funding acquisition, Investigation, Visualization, Methodology, Writing - original draft, Project administration, Writing - review and editing

### Author ORCIDs

Nina Morgner ⓘ https://orcid.org/0000-0002-1872-490X
Benesh Joseph ⓘ https://orcid.org/0000-0003-4968-889X

Reviewer #1 (Public review): https://doi.org/10.7554/eLife.99338.3.sa1
Reviewer #2 (Public review): https://doi.org/10.7554/eLife.99338.3.sa2
Reviewer #3 (Public review): https://doi.org/10.7554/eLife.99338.3.sa3
Author response https://doi.org/10.7554/eLife.99338.3.sa4

## Additional files

### Supplementary files
• MDAR checklist

### Data availability

All the original PELDOR/DEER data are presented in the manuscript. Additionally, data can be downloaded at: https://doi.org/10.5061/dryad.cfxpnvxgd.

The following dataset was generated:

| Author(s) | Year | Dataset title | Dataset URL | Database and Identifier |
|---|---|---|---|---|
| Dajka M, Rath T, Morgner N, Joseph B | 2024 | Dynamic basis of lipopolysaccharide export by LptB2FGC | https://doi.org/10.5061/dryad.cfxpnvxgd | Dryad Digital Repository, 10.5061/dryad.cfxpnvxgd |

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
