## [Editor Report · eLife assessment]

This study provides an **important** advance in the molecular understanding of the lipopolysaccharide export mechanism and machinery in bacteria. By using advanced spectroscopy approaches, the experiments provide **convincing** biophysical support for the dynamic behavior of the multisubunit Lpt transport system. This work has implications for understanding bacterial cell envelope biogenesis and developing drugs that target Gram negative pathogens.

---

## [Referee Report · Reviewer #1 (Public review)]

Summary:

The current manuscript uses electron spin resonance spectroscopy to understand how the dynamic behavior and conformational heterogeneity of the LPS transport system change during substrate transport and in response to the membrane, bound nucleotide (or transition state analog) and accessory subunits. The study builds on prior structural studies to expand our molecular understanding of this highly significant bacterial transport system.

Strengths

This series of well-designed and well-executed experiments provide new mechanistic insights into the dynamic behavior of the LPS transport system. Notable new insights provided by this study include its indication of the spatial organization of the LptC domain, which was poorly resolved in structures, and how the LptC domain modulates the dynamic behavior of the gate through which lipids access the binding site. In addition, a mass spectrometry approach designed to examine LPS binding at different stages in the nucleotide-dependent conformational cycle provides insight into the order of operations of LPS binding and transport.

---

## [Referee Report · Reviewer #2 (Public review)]

Lipopolysaccharide (LPS) is a major component of the outer membrane of Gram-negative bacteria and plays a critical role in bacterial virulence. The LPS export mechanism is a potential target for new antibiotics. Inhibiting this process can render bacteria more susceptible to the host immune system or other antibacterial agents. Given the rise of antibiotic-resistant bacteria, novel targets are urgently needed. The seven LPS transport (Lpt) proteins, A-G, move LPS from the inner to the outer membrane. This study investigated the conformational changes in the LptB2FG-LptC complex using site-directed spin labeling (SDSL) electron paramagnetic resonance (EPR) spectroscopy, revealing how ATP binding and hydrolysis affect the LptF β-jellyroll domain and lateral gates. The findings highlight the role of LptC in regulating LPS entry, ensuring efficient and unidirectional transport across the periplasm.

The β-jellyrolls are not fully resolved in the vanadate-trapped structure of LptB2FG and LptB2FGC. Therefore, the current study provides valuable information on the functional dynamics of these periplasmic domains, their interactions, and their roles in the unidirectional transport of LPS. Additionally, the dynamic perspective of the lateral gates in LptFG in the presence and absence of LptC is another strength of this study. Moreover, at least in detergent samples, more comprehensive intermediates of the ATP turnover cycle are studied than in the available structures, providing crucial missing mechanistic details.

Other major strengths of the study include high-quality DEER/PELDOR distance measurements in both detergent and proteoliposomes, the latter providing valuable dynamics information in the lipid environment. The proteoliposome study is crucial since the previous structural study (Li, Orlando & Liao 2019) was done in rather small-diameter nanodiscs, which might affect the overall dynamics of the complex. It would have been beneficial if the investigators had reconstituted the complex in lipid nanodiscs with the same composition as proteoliposomes. The mixed lipid/detergent micelles provide an alternative. It seems the ATPase activity of the protein complex is much lower in detergent compared with lipid nanodiscs (Li, Orlando & Liao 2019). It is unclear how ATPase activity in proteoliposomes compares to that in detergent micelles.

Additionally, from previous structural studies and the mass spectrometry data presented here, LPS co-purifies and is already bound to the complex, thus the Apo state may represent the LPS-bound state without nucleotides.

---

## [Referee Report · Reviewer #3 (Public review)]

Summary:

The manuscript by Dajka and co-workers reports the application of a biophysical approach to analyse the dynamics of the LptB2FG-C ABC transporter, involved in LPS transport across the cell envelope in *Escherichia coli*. LptB2FG-C belongs to a new class of ABC transporters (type VI) and is essential and conserved in several Gram-negative pathogens. Since LPS is the major component of the outer membrane of the Gram-negative cell and is responsible for the low permeability of this membrane to several antibiotics, a deep understanding of the mechanism and function of the LptB2FG-C transporter is crucial for the development of new drugs targeting Gram-negative pathogens.

Several structural studies have been published so far on the LptB2FG-C transporter, disclosing important aspects of the transport mechanism; nevertheless, lack of resolution of some regions of the individual proteins as well as the dynamic nature of the transport mechanism per se (e.g. the insertion and removal of the TM helix of LptC from the TMDs of the transporter during the LPS transport cycle) has greatly limited the understanding of the mechanism that couples ATP binding and hydrolysis with LPS transport. This knowledge gap could be filled by applying an approach that allows the analysis of dynamic processes. The DEER/PELDOR technique applied in this work fits well with this requirement.

Strengths:

In this study the authors provide some new pieces of information on the LptB2FG-C function and the role of LptC in the transporter using a technique that allowed them to appreciate missing intermediate conformations adopted by the proteins during the transport cycle.

The work is timely and well-conceived. The conclusions of the manuscript are supported by solid data and allow the authors to postulate a dynamic model for the mechanism of translocation of LPS across the inner membrane by the LptB2FGC complex.

---

## [Author Response]

The following is the authors’ response to the original reviews.

**Public Reviews:**

**Reviewer #1:**
Summary:The current manuscript uses electron spin resonance spectroscopy to understand how the dynamic behavior and conformational heterogeneity of the LPS transport system change during substrate transport and in response to the membrane, bound nucleotide (or transition state analog), and accessory subunits. The study builds on prior structural studies to expand our molecular understanding of this highly significant bacterial transport system.StrengthsThis series of well-designed and well-executed experiments provides new mechanistic insights into the dynamic behavior of the LPS transport system. Notable new insights provided by this study include its indication of the spatial organization of the LptC domain, which was poorly resolved in structures, and how the LptC domain modulates the dynamic behavior of the gate through which lipids access the binding site. In addition, a mass spectrometry approach designed to examine LPS binding at different stages in the nucleotide-dependent conformational cycle provides insight into the order of operations of LPS binding and transport.

We thank the reviewer for the very positive comments and highlighting the important findings from our study.

**Reviewer #2 (Public Review):**
Lipopolysaccharide (LPS) is a major component of the outer membrane of Gram-negative bacteria and plays a critical role in bacterial virulence. The LPS export mechanism is a potential target for new antibiotics. Inhibiting this process can render bacteria more susceptible to the host immune system or other antibacterial agents. Given the rise of antibiotic-resistant bacteria, novel targets are urgently needed. The seven LPS transport (Lpt) proteins, A-G, move LPS from the inner to the outer membrane. This study investigated the conformational changes in the LptB2FG-LptC complex using site-directed spin labeling (SDSL) electron paramagnetic resonance (EPR) spectroscopy, revealing how ATP binding and hydrolysis affect the LptF βjellyroll domain and lateral gates. The findings highlight the role of LptC in regulating LPS entry, ensuring efficient and unidirectional transport across the periplasm.The β-jellyrolls are not fully resolved in the vanadate-trapped structure of LptB2FG and LptB2FGC. Therefore, the current study provides valuable information on the functional dynamics of these periplasmic domains, their interactions, and their roles in the unidirectional transport of LPS. Additionally, the dynamic perspective of the lateral gates in LptFG in the presence and absence of LptC is another strength of this study. Moreover, at least in detergent samples, more comprehensive intermediates of the ATP turnover cycle are studied than in the available structures, providing crucial missing mechanistic details.

We thank the reviewer for highlighting our major findings!

Other major strengths of the study include high-quality DEER distance measurements in both detergent and proteoliposomes, the latter providing valuable dynamics information in the lipid environment. However, lipid composition is not mentioned. The proteoliposome study is crucial since the previous structural study (Li, Orlando & Liao 2019) was done in rather small-diameter nanodiscs, which might affect the overall dynamics of the complex. It would have been beneficial if the investigators had reconstituted the complex in lipid nanodiscs with the same composition as proteoliposomes. The mixed lipid/detergent micelles provide an alternative. It seems the ATPase activity of the protein complex is much lower in detergent compared with lipid nanodiscs (Li, Orlando & Liao 2019). In the current study, ATPase activity in proteoliposomes is not provided. Also, the reviewer assumes cysteine-less (CL) constructs of the complex components were utilized. The ATPase assay on CL complex is not presented. Additionally, from previous structural studies and the mass spectrometry data presented here, LPS co-purifies and is already bound to the complex, thus the Apo state may represent the LPS-bound state without nucleotides.

The liposomes are made from *E. coli* polar lipid extract, which we added to the Materials and Methods part now. We could not yet perform the investigations in nanodiscs, which is one of our aims for future. The ATPase activity is lower in micelles and the reviewer is correct in that we did not perform/compare ATPase activity in proteoliposomes. The data denoted as wild-type (WT, Figure S4) corresponds to the cysteine-less (CL) variant, which is now corrected in the supporting information. As the reviewer commented, the mass spectrometry data reveal bound LPS in the apo-state. However, as seen from our results, ADP-Mg2+ state is similar to the apo state, thus in the cellular environment LPS may bind to this state as well.

The selection of sites to probe lateral gate 2, which forms the main LPS entry site, may pose an issue. Although the authors provide justification based on the available structures, one site (position 325 in LptF) is located on a flexible loop, and position 52 in LptG is on the neighboring transmembrane helix, separated by a potentially flexible loop from the gating TM1. These labeling sites could exhibit significant local dynamics, resulting in a broader distribution of distances and potentially masking the gating-related conformational changes.

Position 52 in LptG is located at the beginning of the neighboring transmembrane helix. As we have discussed in the manuscript, position 325 in LptF is located on a short loop connected to TM5. In the structures, this loop shows a very similar orientation (Figure S6). Further, the observed heterogeneity for the lateral gate-2 is considerably modulated into distinct conformation(s) upon LptC binding (Figure 6D-E). This would not be the case if this loop possesses any independent flexibility. Confirming these observations, the room temperature continuous wave ESR spectra revealed the least flexibility for this spin pair (Figure S5, S7). In view of the reasons and observations detailed above, we conclude that the local flexibility at the labelled sites might not make any significant contribution for the broad distribution observed at this gate in LptB2FG (Figure 4).

**Reviewer #3 (Public Review):**
Summary:The manuscript by Dajka and co-workers reports the application of a biophysical approach to analyse the dynamics of the LptB2FG-C ABC transporter, involved in LPS transport across the cell envelope in *Escherichia coli*. LptB2FG-C belongs to a new class of ABC transporters (type VI) and is essential and conserved in several Gram-negative pathogens. Since LPS is the major component of the outer membrane of the Gram-negative cell and is responsible for the low permeability of this membrane to several antibiotics, a deep understanding of the mechanism and function of the LptB2FG-C transporter is crucial for the development of new drugs targeting Gram-negative pathogens.Several structural studies have been published so far on the LptB2FG-C transporter, disclosing important aspects of the transport mechanism; nevertheless, lack of resolution of some regions of the individual proteins as well as the dynamic nature of the transport mechanism per se (e.g. the insertion and removal of the TM helix of LptC from the TMDs of the transporter during the LPS transport cycle) has greatly limited the understanding of the mechanism that couples ATP binding and hydrolysis with LPS transport. This knowledge gap could be filled by applying an approach that allows the analysis of dynamic processes. The DEER/PELDOR technique applied in this work fits well with this requirement.Strengths:In this study, the authors provide some new pieces of information on the LptB2FG-C function and the role of LptC in the transporter. Notably, they show that:- There is high heterogeneity in the conformational states of the entry gate of LPS in the transporter (gate-2) that are reduced by the insertion of LptC, and the heterogeneity observed is not altered by ATP binding or hydrolysis (as expected since LPS entry is ATP-independent).- ATP binding induces an allosteric opening of LptF β-jellyroll domain that allows for LPS passage to the β-jellyroll of LptC, which is stably associated with the β-jellyroll of LptF throughout the cycle.- The β-jellyroll of LptG is highly flexible, indicating an involvement in the LPS transport cycle.The manuscript is timely and overall clear.

We thank the reviewer for the positive comments and highlighting our findings and the strength of DEER/PELDOR spectroscopy for characterizing the dynamics aspect of the LPS transport system.

Weaknesses:I list my concerns below and provide suggestions that, in my opinion, should be addressed to reinforce the findings of this study.(1) Protein complex controls: the authors assess the ATPase activity of the spin-labelled variants of their protein complexes to rule out the possibility that engineering the proteins to enable spin labelling could affect their functionality (Figure S4). It has been reported that the association of LptC to LptB2FG complex inhibits its ATPase activity. However, in the ATPase assay data shown in Figure S4, the inhibitory effect of the LptC TM is not visible (please compare LptB2FG F-A45C G-I335C and F-L325C G-A52C with and without LptC). This can lead to suspect that the regulatory function of LptC is missing in the LptC-containing complexes used in this work. I suggest the authors include wt LptB2FGC in the assay to compare the ATPase activity of this complex with wt LptB2FG. The published inhibitory effect of TM LptC has been observed in proteoliposomes. Since it is not clear from the paper if the ATPase assay in Figure 4 has been conducted in DDM or proteoliposomes, the lack of inhibitory effect could be due to the assay conditions. A comparative test could answer this question.

We could not observe the inhibitory effect of LptC on the ATPase activity of LptB2FG. As the reviewer pointed out, the primary reason is that we performed the assays in detergent micelles and not in proteoliposomes. For this reason, a comparison of the activity between (cysteine-less) LptB2FG and LptB2FG-C as the reviewer suggested would not be informative. As this information is not directly relevant for our current interpretations, we plan to perform those experiments in liposomes in the near future.

(2) Figure 2: NBD closure upon ATP binding to LptB2FG is convincingly demonstrated both in DDM micelles and proteoliposomes, validating the experimental system. However, since under physiological conditions, ATP binding should take place before the displacement of the TM of LptC (Wilson and Ruiz, Mol microbiol 2022), I suggest the authors carry out the experiments with LptC-containing complexes to investigate conformational changes (if any) that are triggered when ATP binding occurs before the TM displacement.

We thank the reviewer for the suggestion. These experiments are in our to do list and would be performed in the near future.

(3) Proteoliposomes: in the experiments shown in Figures 3 and 4, unlike those in Figure 2, measurements in proteoliposomes give different results from the experiments in DDM, showing higher heterogeneity. Could this be related to the presence (or absence) of LPS in liposomes? It is not mentioned in the materials and methods section whether LPS is present. Could the authors please discuss this?

We thank the reviewer for bringing out this interesting point. The liposomes are made from *E. coli* polar lipid extract. In the polar lipid extract, phosphatidylethanolamine (PE) is the predominant lipid component with minor amounts of phosphatidylglycerol (PG) and cardiolipin. Thus, the differences in the heterogeneity we observed in proteoliposomes might not be due to the presence of LPS. We added a short description on this aspect in the ‘Discussion’ part.

(4) The authors show large conformational heterogeneity in gate-2 (using the spin-labelled pair F-L325R1-G-A52R1) and suggest that deviation from the corresponding simulations could be due to the need for enhanced dynamics to allow for gate interaction with LPS or LptC. The effect of LptC is probed in the experiments shown in Figure 6, but I suggest the authors add LPS to the complexes to evaluate the possible stabilizing effect of LPS on the conformations shown in Figure 4.

This indeed is an important experiment, which we plan to do in the near future.

(5) Figure 6: the measurement of lateral gate 1 and 2 dynamics in the LptC-containing complexes clearly supports the hypothesis, proposed based on the available structures, that TM LptC dissociates from LptB2FG upon ATP binding. However, direct evidence of this movement is still missing. Would it be possible to monitor the dynamics of the TM LptC by directly labelling this protein domain? This would give a conclusive demonstration of the displacement during the ATPase cycle.

Yes, it should be possible to label LptC and monitor its position with respect to LptF or LptG. These experiments are in progress in our laboratory.

(6) LPS release assay: Figure 6 panels H-I-J show the MS spectra relative to LPS-bound and free proteins obtained from wt LptB2FG upon ATP binding and ATP hydrolysis conditions. From these spectra the authors conclude that LPS is completely released only upon ATP hydrolysis. However, the current model predicts that LPS release into the Lpt bridge made by LptC-A-D is triggered by ATP binding. For this reason, I suggest the authors assess LPS release also from the LptB2FGC complex where, in the absence of LptA, LPS would be expected to be mostly retained by the complex under the same conditions.

These indeed are exciting experiments. LPS binding and release by LptB2FGC is in progress in our laboratories.

**Recommendations for the authors:**

**Reviewer #1 (Recommendations For The Authors):**
Page 2 typo: apo-sate should be apo-state

Thank you! We corrected the typo.

Can the authors clarify whether LPS is co-purified with the protein? Does it remain bound throughout the liposome reconstitution process?

Our mass spectrometry data show that LPS is co-purified with LptB2FG in micelles. However, we cannot yet verify the presence of bound LPS after reconstitution into proteoliposomes. We added a sentence in the last paragraph before Discussion as ‘Thus, LPS is co-purified with LptB2FG in micelles.’

**Reviewer #2 (Recommendations for The Authors):**
Several points require clarification:(1) The reviewer would have benefited from access to the raw DEER traces. For instance, in Figure 4, the change in the raw data appears subtle. The differences between the Apo and vanadate-trapped states in b-DDM might be related to a lower signal-to-noise ratio in the Apo state.

We would be happy to share the raw DEER data upon request. The analysis is performed with the primary data, which also takes into account of the noise level for the calculating the confidence interval. Therefore, the distances with the 95% confidence interval are reliable to the extent as they are presented.

(2) The panel labels in Figures 2-4 do not match the legends.

Thank you! We corrected them.

(3) In Figure 2G, the authors state, "Overall, the ATP-induced closure as observed in micelles (and the structures) is maintained in the native-like lipid bilayers for the NBDs." This statement is technically incorrect since the vanadate-trapped state is not equivalent to the ATP+EDTA "ATP binding" state, which was not tested in proteoliposomes (PLS). The authors should have tested this condition for a few mutants in proteoliposomes. They should revise the manuscript to reflect this or provide evidence that the ATP+EDTA state is similar to the vanadate-trapped state in PLS.

We corrected the sentence as ‘Overall, the nucleotide-induced closure as observed in micelles (and the structures) is maintained in the native-like lipid bilayers for the NBDs.’

(4) The mutant F-L325R1_G-A52R1 is not optimal for probing gate 2. Specifically, position 325 in LptF is highly flexible, as indicated by the very broad distance distributions in Figure 4, and may hinder probing the associated conformational changes in this gate. Comparing the cryo-EM structures of this loop under different conditions (Figure S6) does not provide solid evidence for the lack of flexibility.

Position 52 in LptG is located at the beginning of the neighboring transmembrane helix. As we have discussed in the manuscript, position 325 in LptF is located on a short loop connected to TM5. In the structures, this loop shows a very similar orientation (Figure S6). Further, the observed heterogeneity for the lateral gate-2 is considerably modulated into distinct conformation(s) upon LptC binding (Figure 6D-E). This would not be the case if this loop possesses any independent flexibility. Confirming these observations, the room temperature continuous wave ESR spectra revealed the least flexibility for this spin pair (Figure S5, S7). In view of the reasons and observations detailed above, we conclude that the local flexibility of the labelled sites might not make any significant contribution for the broad distribution observed at this gate in LptB2FG (Figure 4).

(5) Regarding Figure 4B, the authors state, "In the vanadate-trapped and ATP samples, the major population is centered at 2 nm (which corresponds to the simulation on the vanadate trapped structure)". While the shift to shorter distances aligns with the structures, the average distance from the simulation is around 3 nm and does not correspond closely to the DEER distances of 2 nm.

Thank you for noting this point. We corrected the sentence as ‘In the vanadate-trapped and ATP samples, the major population is centred at 2 nm (which is closer to the simulation on the vanadate-trapped structure).’

(6) Regarding Figure 4D, the authors state, "Unlike the lateral gate-1 (and the NBDs), ADP-Mg2+ also induced a similar shift in the distance distribution." The reviewer believes that even without interaction with LptC, an equilibrium exists between two states in gate-2, and ATP binding or vanadate-trapping shifts the equilibrium to a shorter-distance population. Additionally, if the signal-to-noise ratio of the Apo state were similar to that of the ADP-Mg2+ state, similar distance distributions would have been observed for the Apo state.

We thank the reviewer for bringing out this excellent point. We thoroughly modified the corresponding section as ‘ADP-Mg2+ also gave a broad distribution comparable to the apo-state. Thus, in the apo-state this gate appears to exist in an equilibrium between the two conformations observed from the corresponding structures. ATP binding or vanadate-trapping shifts the equilibrium towards the collapsed conformation.’

(7) Defining the conformational dynamics of the b-jellyroll domains is one of the major strengths of this study. The LptF and LptG b-jellyroll domains exhibit high flexibility in detergent micelles. Unfortunately, none of the experiments were repeated in proteoliposomes to determine if this flexibility persists in a lipid environment.

As it is conceivable, it is truly beyond the scope of the current study to repeat all the measurements in liposomes. Currently we are extending those investigations to liposomes and would be able to provide more insights in the near future.

(8) Regarding Figure 6G, the authors claim, "Distances corresponding to the apo state are present possibly due to an incomplete vanadate trapping for this sample." It is unlikely that vanadate trapping would be incomplete for just one sample. A repeat experiment is recommended.

We will update on this point is due time.

(9) Regarding the structural dynamics of the lateral gates, detergent micelles, and liposomes are vastly different environments. It is challenging to reach a consensus model based on data mostly derived from detergent micelles and only a few from proteoliposomes.

The observations in PLS are qualitatively similar to the micellar sample for the investigated positions (please see the first paragraph in “Discussion”). Further, our observations are in agreement with previous structural and biochemical data and further extent the mechanism in a coherent manner.

**Reviewer #3 (Recommendations For The Authors):**
Minor comments(1) Figure legends: There are several mismatches between panel nomenclature and the corresponding descriptions in the legends. Please check the correspondence between panel identification and descriptions throughout the manuscript (for example, F-G and H-J in Figure 2; and I and H in Figure 3).

Thank you! We corrected them.

- Figure 6 legend: asterisk is in panel D and not C.

Corrected

- Panels E and F are not mentioned. Moreover, the spectra for vanadate trapped conformation of LptF219-LptC104 have not been given a letter.

Corrected

- A description of the different colors in the "Distance r" axis should be added to figure 2, 3, and 4 legends.

Corrected

- Please indicate the meaning of the black arrows in figure legends.

Corrected

(2) To improve data comprehension by the readers, the authors should indicate the relative spinlabelled pairs on the top of Figure 2, 3, and 4, as done for Figures 5 and 6.

Done

(3) Reference 56 is cited incorrectly in the reference list and refers to a study employing reconstituted LptB2FG complexes rather than isolated β-jellyroll domains.

Corrected

(4) Figure 3: How do the authors explain the evidence that ATP binding influences gate 1 conformational flexibility only in DDM micelles with respect of PLS? Is this something related to the release of LPS from the complex in different environments?

We do not know whether this difference is related to LPS release. Therefore, we generally interpreted as an effect of the membrane environment.

(5) The initial sentence of the discussion looks somewhat incomplete, please correct it.

Done

(6) To improve the readability of the paper, it could be useful to better focus the topic of the headings of the result paragraphs concerning the analysis of the individual lateral gates (for example, by indicating the name of the gate in the headings).

Done